# Anticancer Properties of Macroalgae: A Comprehensive Review

**DOI:** 10.3390/md23020070

**Published:** 2025-02-07

**Authors:** Sara Frazzini, Luciana Rossi

**Affiliations:** Department of Veterinary Medicine and Animal Sciences—DIVAS, University of Milan, via dell’Università 6, 26900 Lodi, Italy; luciana.rossi@unimi.it

**Keywords:** macroalgae, seaweed, cancer, anticancer, anticarcinogenic, bioactive compounds, fucoidans, phlorotannins, terpenoids, immunotherapy

## Abstract

In recent years, the exploration of bioactive molecules derived from natural sources has gained interest in several application fields. Among these, macroalgae have garnered significant attention due to their functional properties, which make them interesting in therapeutic applications, including cancer treatment. Cancer constitutes a significant global health burden, and the side effects of existing treatment modalities underscore the necessity for the exploration of novel therapeutic models that, in line with the goal of reducing drug treatments, take advantage of natural compounds. This review explores the anticancer properties of macroalgae, focusing on their bioactive compounds and mechanisms of action. The key findings suggest that macroalgae possess a rich array of bioactive compounds, including polysaccharides (e.g., fucoidans and alginates), polyphenols (e.g., phlorotannins), and terpenoids, which exhibit diverse anticancer activities, such as the inhibition of cell proliferation, angiogenesis, induction of apoptosis, and modulation of the immune system. This review provides an overview of the current understanding of macroalgae’s anticancer potential, highlighting the most promising compounds and their mechanisms of action. While preclinical studies have shown promising results, further research is necessary to translate these findings into effective clinical applications.

## 1. Introduction

Algae, the most primitive group of plants, are simple photosynthetic organisms that have been thriving on Earth for an astonishing 3.5 billion years [1]. This longevity is a testament to their resilience and adaptability. Over the course of their existence, these remarkable organisms have evolved, giving rise to two distinct types: macroalgae, also known as seaweed, and microalgae [2,3]. These diverse forms of algae play a crucial role as primary producers, kickstarting the aquatic food chain and providing sustenance for a wide array of organisms, including fish, crustaceans, and gastropods [4].

Algae, an incredibly diverse group of organisms, encompass a wide range of simple, typically autotrophic life forms [5]. Just like plants, algae are capable of photosynthesis, harnessing the power of sunlight to convert carbon dioxide and water into energy-rich nutrients [6,7]. Algae hold immense ecological significance in the marine biome, playing a pivotal role in the survival and well-being of countless organisms. They perform a vital function in global oxygen production and are responsible for generating approximately 70% of the Earth’s oxygen [8]. This staggering contribution underscores the urgent need to preserve and protect these organisms. In recent years, there has been a growing interest in the use of algae in various industrial and research fields. The reason for this keen fascination lies in the findings of marine biologists who have successfully identified an array of over 10,000 algae-produced bioactive compounds [9,10,11]. These biochemical compounds are recognized for their functional properties, which are capable of producing various biological effects, such as antioxidant or antimicrobial activity [12]. Given the diverse effects that these bioactive compounds exhibit, algae play a significant role in functional nutrition, offering a diverse array of health benefits that are increasingly recognized in dietary practices. Marine algae, such as brown and blue-green algae, have been identified as potential sources of anticancer agents [13,14]. Research has shown that compounds isolated from marine algae can effectively target gastrointestinal cancers, such as stomach and colon cancer, which are among the most prevalent forms of cancer globally [15,16]. The bioactive compounds derived from algae have been reported to not only prevent cancer development but also enhance the efficacy of conventional treatments. For instance, these compounds can modulate cancer cell metabolism and influence the tumor microenvironment, thereby reducing tumorigenesis and preventing metastasis [17,18]. Furthermore, the antioxidant and anti-inflammatory properties of algal components contribute to their protective effects against cancer [16]. As research continues to uncover the diverse mechanisms through which these bioactive compounds operate, the integration of algae into functional foods and nutraceuticals is becoming increasingly relevant for cancer prevention and treatment strategies [15,17]. Thus, algae represent a promising avenue for enhancing health outcomes in cancer care.

Cancer represents the second leading cause of death worldwide since, according to the latest global cancer burden data released by the World Health Organization (WHO)’s International Agency for Research on Cancer (IARC), 20 million new cancer cases were diagnosed worldwide in 2020, with mortality equal to 9 million of people worldwide [19].

Research has been focused on the development of new therapeutic strategies to extend life expectancy and reduce cancer mortality. Although surgical resection, radiotherapy, and chemotherapy remain the conventional treatment strategies, they often lead to drug resistance and unpleasant side effects [20,21]. In recent years, new anticancer therapies have emerged, and in this context, natural products, such as algae, have gained attention [22]. Thereby, the hypothesis that bioactive compounds in algae may have an antitumor function has direct parallels with the contemporary era of chemotherapy, which seeks to exclusively target and eliminate cancer cells while minimizing damage to healthy cells.

## 2. Macroalgae: A Comprehensive Overview

Algae are a notable group of non-flowering organisms that resemble plants, and they share the presence of chlorophyll and the ability to carry out photosynthesis. Unlike plants, they do not have typical stems, leaves, and roots, nor do they possess vascular tissue. Algae can be made by single cells or colonies or can be multicellular organisms [23]. They can be found abundantly in oceans, ponds, and lakes and have various practical uses, including providing food for both humans and animals and producing natural plant chemicals, pharmaceuticals, cosmetics, and industrial products [24,25]. Algae also have a vital role in the environment as they help purify it by removing excessive nutrients and CO_2_ [26,27]. Due to their vast array of morphological, anatomical, and reproductive characteristics, as well as the various types of photosynthetic pigments and cell wall composition, algae become difficult to classify. The main basis for classification lies in their photosynthetic pigments and major cell products. This leads to the subdivision of the plant kingdom into numerous phyla, where algae are grouped either independently or alongside other less complex plants. Within these phyla, ten distinct categories can be identified, with some comprising only one algae class and others consisting of multiple classes. In the first instance, algae are differentiated between macro- and microalgae [28,29,30]. Macroalgae, also known as seaweeds, are generally found in the ocean, representing an important biological resource as well as an alternative food source [31], given their high protein, dietary fiber, vitamin, and mineral content [32,33]. Nowadays, macroalgae are taxonomically classified based on the nature of their pigments. Following this classification, weeds are divided into red (Rhodophyta), brown (Phaeophyta), and green (Chlorophyta) (Figure 1) [34]. Red macroalgae belong mainly to the maritime environment, constituting a unique group distinguished by their eukaryotic cells devoid of flagella and centrioles. Their chloroplasts lack external endoplasmic reticulum and contain unstacked thylakoids known as stroma. Additionally, they utilize phycobiliproteins as accessory pigments, imparting their characteristic red hue. Their red pigment allows them to photosynthesize at deeper depths than other algae. Similarly, another element that distinguishes this category of algae from others is that red algae are able to store carbohydrates as Floridean starch outside the chloroplasts [35,36,37]. Brown macroalgae, exclusively multicellular organisms, are the second largest group of macroscopic algae, with around 2000 identified species worldwide, mainly located in the colder waters of the Northern Hemisphere [38]. Brown algae produce a large amount of carotenoid, among which is fucoxanthin, the pigment responsible for their distinctive greenish-brown color [39,40]. Green algae exhibit great diversity of form and function. Like red algae, they can be unicellular, multicellular, colonial, or coenocytic. They have membrane-bound chloroplasts and nuclei. Most green algae are aquatic and found commonly in freshwater (mainly charophytes) and marine habitats (mostly chlorophytes); some are terrestrial, growing on soil, trees, or rocks (mostly trebouxiophytes) [41,42,43].

## 3. Macroalgae’s Bioactive Compounds with Anticancer and Anticarcinogenic Potential

Nowadays, there is an increasing interest in algae. This is because, in addition to their nutritional qualities, they are a rich source of many biologically active compounds and one of the richest sources of natural bioactive substances, such as polysaccharides, proteins, lipids, polyphenols, carotenoids, pigments, vitamins, sterols, and enzymes [44,45]. All these compounds confer to the algae different functional properties, such as antioxidant, antibacterial, antiviral, and antifungal [46,47,48]. Among all the possible potentiality of algae in the last year, scientific research has delved deeper into the realm of algae, developing a growing awareness regarding their potential health benefits, particularly in terms of cancer prevention and tumor regression [49]. In fact, it was recently demonstrated that the bioactive compounds of seaweeds could exhibit cytotoxic effects on cancer cells, inhibiting tumor growth through apoptosis induction and interference with kinases and cell cycle pathways, safeguarding cells from DNA damage, reducing chronic inflammation, and contributing to the prevention and regression of tumors [50,51,52,53,54]. According to their different categories, algae species are characterized by different bioactive compounds (Figure 2; Table 1).

Green algae, due to their diversity, offer different anticancer bioactive compounds, which possess potent antioxidant properties and exhibit significant antitumor activities [77]. Their ability to inhibit tumor growth, reduce DNA damage, and block the formation of mutagenic agents makes them promising candidates for the development of novel anticancer therapies and preventive strategies. Additionally, the antioxidant properties of these bioactive compounds play a crucial role in protecting cells from oxidative damage, which is a major contributor to the development of cancer. By neutralizing harmful free radicals, the green pigments found in algae can help reduce the risk of DNA mutations and subsequent tumor formation [78,79,80]. Going into detail about the different bioactive compounds characterizing green algae, chlorophylls, such as chlorophyll a and b, are magnesium-rich pigments that have been found to possess antitumor, antigenotoxic, and antimitotic properties. These compounds have shown promising results in reducing liver tumors induced by aflatoxin B1 and other chemical carcinogens in experimental animal models. They have also demonstrated the ability to decrease DNA damage and inhibit the formation of carcinogen-derived DNA adducts, which are mutagenic agents. As well as chlorophylls, their derivatives, such as tetraprenyltol and siphonaxanthin, disclose anticancer activity [59,60,61,81]. In particular, siphonaxanthin has shown remarkable cytotoxicity against human colon cancer cell lines, demonstrating its potential as a powerful weapon against this disease [82]. Xanthophylls, including lutein and zeaxanthin, are another group of green pigments that exhibit chemopreventive effects against oral and liver cancer [83]. Dietary intake of these compounds has been associated with a decreased incidence of tumors and preneoplastic lesions, particularly in the liver. Finally, β-carotene was discovered to be capable of converting into retinol and stimulating the induction of retinol-binding protein in the body, which, in turn, plays a vital role in the prevention and mitigation of cancer [21]. In the realm of red algae, until now, only a few species have been studied for their natural products and biological activities. To date, bioactive compounds with undetected anticancer properties belong mainly to the classes of polysaccharides, secondary metabolites, and halogenated compounds [84]. Specifically, sulfated polysaccharides, such as carrageenans and agarans, have been identified as potent anticancer agents. Their effectiveness lies in their immunomodulatory effects, which have been shown to inhibit cancer cell growth and metastasis. These polysaccharides act as warriors, fighting against cancer’s destructive tendencies and offering hope for improved patient outcomes [85]. Guaiane sesquiterpenoids are a type of secondary metabolite produced by plants, which have been found to possess anticancer properties. This compound exists in the form of sesquiterpenoid alcohol and has shown promising results in both in vitro and in vivo tests on leukemia L1210 cells where it functions as an expellant by inducing apoptosis in the early stages of polynucleosome formation [86,87]. Moreover, red algae are a valuable source of halogenated compounds, including bromide, bromoform, and bromophenols, which are recognized as potential anticancer agents. Unfortunately, due to their limited availability and negative side effects, further exploration of these compounds is necessary to better clarify their role in cancer therapy [88,89,90,91]. Turning our attention to brown algae, renowned for their rich bioactive content, they have also exhibited promising anticancer activity [92,93]. Among the bioactive substances found in brown algae, fucoxanthin, a xanthophyll, has garnered significant attention. Information gathered from various research studies suggests that fucoxanthin can hinder the development of various forms of tumors and impede their spread to other parts of the body [94,95,96].

## 4. Mechanisms of Action

The anticancer effect of algae-derived compounds occurs through multiple mechanisms of action (Figure 3).

One strategy is to induce apoptosis. Algal compounds, such as phlorotannins from brown algae and fucoxanthin, are shown to upregulate apoptosis-associated proteins [22,97], including caspase activation, which are crucial enzymes in the apoptosis process, and modulate mitochondrial pathways and death receptor signaling, leading to the effective elimination of cancer cells [98,99]. Additionally, seaweed-derived polyphenols and other bioactive compounds interfere with cell cycle progression, arresting cancer cell growth at various stages and preventing proliferation [51]. For instance, compounds from *Chlorella* sp. have been found to inhibit the AKT/mTOR survival signaling pathway, which is essential for cell growth and survival, thereby inducing apoptosis in cancer cells [100]. Among bioactive algal compounds with apoptotic activity, Eo et al. highlight that Phlorofucofuroeckol A is able to induce apoptosis by interfering with the pathway of AFT3, a major regulator of metabolic homeostasis that can influence cancer cell proliferation [101]. On the other hand, Zhangfan et al. disclosed that fucosterol extract could target the Raf/MEK/ERK signaling pathway, which, given its involvement in the proliferation and tumorigenesis of several cancer types, is to be considered an essential therapeutic target [102,103]. Moreover, marine-derived compounds, such as polysaccharides, peptides, and terpenoids, exhibit cytotoxic effects on cancer cells by disrupting key signaling networks, like the PI3K/AKT, ROS, and p53 pathways, which are involved in cell survival, oxidative stress response, and DNA repair mechanisms [104]. The anti-angiogenic and anti-metastatic properties of these compounds further contribute to their anticancer efficacy by inhibiting the formation of new blood vessels and the spread of cancer cells to other parts of the body [105]. The possible molecular mechanisms are attributed to inhibiting the signal transduction by the angiogenic factor VEGF (Vascular Endothelial Growth Factor) and migration of cancer cells. Additionally, these algae-derived compounds possess the ability to modulate different signaling pathways that regulate cell survival, proliferation, and metastasis. These bioactive components exert their effects by interacting with specific targets involved in tumor progression, such as growth factor receptors, cell adhesion molecules, and transcription factors [106,107]. Additionally, studies have revealed that algae-derived compounds, including polyphenols, carotenoids, lipids, and polysaccharides, can modulate the tumor microenvironment, influencing critical factors such as inflammation, immune response, and angiogenesis. By targeting these components, they contribute to creating an unfavorable environment for tumor growth and progression. Furthermore, these compounds possess antioxidant properties, effectively neutralizing harmful free radicals and reducing oxidative stress, which is known to promote tumor development. Furthermore, the inclusion of these alga-derived bioactive compounds in cancer treatment approaches has displayed promising therapeutic outcomes. They have been shown to enhance the effectiveness of traditional chemotherapy drugs, reducing their toxic side effects while increasing their anticancer activity. Moreover, compounds like fucoidan, phlorotannins, laminarin, carrageenan, and ulvans have demonstrated the potential to sensitize cancer cells to radiation therapy, improving its efficacy in eradicating tumor cells and preventing recurrence [108,109,110]. The diverse chemical structures of these marine bioactive compounds allow them to target multiple pathways simultaneously, making them effective against various cancer types and potentially overcoming drug resistance [111,112]. Furthermore, the presence of specific bioactive compounds, like gallic acid and lutein, in algal extracts has been confirmed to reduce cancer cell viability significantly, highlighting their potential as therapeutic agents [100]. Overall, the multifaceted mechanisms of action of algae and algae-derived compounds, including apoptosis induction, cell cycle arrest, and modulation of critical signaling pathways, underscore their promising role in cancer therapy and warrant further clinical studies to validate their efficacy and safety.

## 5. In Vitro Studies

In vitro studies have extensively assessed the anticancer activity of various algae, revealing promising results across different species of macroalgae (Table 2). Marine algae extracts have been tested for their potential inhibitory or toxic effects on different cell lines. This testing is important in screening the therapeutic and toxic effects of new algal compounds. Growth inhibition is often used as a measure of the extract’s potential anticancer activity. To determine this, cell count, protein synthesis, MTT reduction, and cell cycle monitoring are employed. Marine macroalgae, such as *Gelidiella acerosa* (Rhodophyta), have shown significant antiproliferative activity against A549 lung cancer cells, with ethyl acetate extracts inducing apoptosis through the activation of caspase 3 and altering the Bax:Bcl2 ratio [113]. Furthermore, the butanolic extracts of *Gracillaria corticata* (Rhodophyta) have demonstrated greater anticancer activity against HeLa, K-562, and MDA-MB cell lines compared to other solvent extracts [114]. Green algae, such as *Caulerpa lentillifera* (Chlorophyta), have also exhibited wide-ranging anticancer effects on colorectal, hepatoma, breast cancer cell lines, and leukemia, attributed to their sulfated polysaccharides [115]. Additionally, the methanolic extracts of *Caulerpa taxifolia*, *Ulothrix flacca*, and *Ulva fasciata* (formerly *Ulva fasciata*) (Chlorophyta) have shown ROS-dependent mitochondrial damage-induced cytotoxicity in neuroblastoma cells [116]. Antarctic seaweeds, like *Pyropia endiviifolia* (Rhodophyta), have displayed selective antitumor activity against glioma and lung cancer cells, with extracts reducing cell viability significantly without affecting non-transformed cells [117]. Additionally, it was shown that lipoxygenase products can induce p53-independent apoptosis in tumor cells. These findings, which have sparked a growing interest in using marine algal compounds for cancer treatment, are a testament to the promising results of the in vitro testing of algal extracts and isolated compounds on various cancer cell lines.

In vitro studies remain a crucial step in the preclinical evaluation of macroalgae-derived compounds for cancer treatment, serving as an indispensable first step in identifying bioactive compounds, elucidating their mechanisms of action, and assessing cytotoxicity against cancer cells. Therefore, understanding the mechanisms behind their anticancer effects paves the way for future research, potentially leading to the development of effective and safe chemotherapeutic agents derived from algae [118,119,120].

**Table 2 marinedrugs-23-00070-t002:** In vitro studies reporting the anticancer potential of brown, red, and green macroalgae.

Algal Species (Compounds)	Cell Line	Effects	Reference
Brown Macroalgae			
*Gongolaria usneoides* (formerly *Cystoseira usneoides*) (Phaeophyceae) (meroterpenoids isolate)	HT-29	↓ in ERK/JNK/AKT signaling pathways	[121]
*Dictyota ciliolata* (Phaeophyceae) (Aqueous and methanol extract)	HCT116; HepG2	Cytotoxic effect ↑ Activity of caspase 3 and 9 in HepG2	[122]
*Dictyota cervicornis* (formerly *Dictyota Indica*) (Phaeophyceae) (fucoxanthin)	MDA-MB-231	Fucoxanthin extract has anticancer activity without toxic effects to the normal cells	[123]
*Ecklonia cava* (Phaeophyceae) (Dieckol, commercial Sigma-Aldrich, St. Louis, MO, USA)	PANC-1	Induction of apoptosis and inhibition of the progression of PANC-1 cell lines	[124]
*Ecklonia cava* (Phaeophyceae) (Fucosterol, commercial Sigma-Aldrich, St. Louis, MO, USA)	HCC827; A549; SK-LU-1; A427	Growth inhibition Induction of apoptosis and cell cycle arrest	[103]
*Fucus vesiculosus* (Phaeophyceae) (acetonic extract)	PANC-1; PancTu1; Panc89; Colo357	↓ Cell viability ↑ Apoptotic effects	[125]
*Fucus vesiculosus* (Phaeophyceae) (Fucoidan, commercial Sigma-Aldrich, St. Louis, MO, USA)	A549; CL1-5	Induction of apoptosis cell death	[126]
*Padina pavonica* (Phaeophyceae) (methanol extract)	HepG2	↓ Number and viability of cells	[127]
*Sargassum incisifolium* (formerly *Sargassum heterophyllum*) (Phaeophyceae) (tetraprenylquinone, and sargaquinoic acid)	MCF-7; MDA-MB-231	Apoptosis induction via caspase-3, -6, -8, -9, and -13 ↓ in Bcl-2 gene Arrest of G1 phase in MDA-MB-231 cells	[128]
*Sargassum ilicifolium* (Phaeophyceae) (alcoholic extract)	MCF-7; MDA-MB-231; HeLa; HepG2; HT-29	All the extracts were antiproliferative against all the cancer cell lines, dose-dependently, with the *G. corticata* methanol extract showing the greatest inhibition activity against the MCF-7 cell line	[129]
*Sargassum* sp. (Phaeophyceae) (Ethanol fraction)	MCF-7	Ethanol fraction induced cell shrinkage, cell membrane blebbing, and formation of apoptotic bodies	[130]
*Sargassum wightii* (Phaeophyceae) (Methanol extract)	HT-29	Inhibition of the proliferation	[131]
*Undaria piannatifida* (Phaeophyceae) (fucoxanthin, commercial Wuhan Heli)	MDA-MB-231	↓ in growth and cell proliferation	[132]
Green Macroalgae			
*Botryidiopsidaceae* sp. (Chlorophyta) (ethanol extract)	HeLa; HCT116	↑ Expression of pro-apoptotic gene (p53) ↓ Expression of anti-apoptotic gene (Bcl-2)	[133]
*Caulerpa lentillifera* (Chlorophyta) (sulfated polysaccharide)	HCT-8; HL-60; K-562; KG-1a; MCF-7; MDA-MB-231	Cytotoxic effect	[115]
*Caulerpa racemosa* (Chlorophyta) (methanol extract)	HL-60	↓ Cancer cell growth Apoptotic body formation	[134]
*Caulerpa racemosa* (Chlorophyta) (chloroform extract)	KAIMRC1	Chloroform fraction and crude polyphenolic extract exhibited moderate cytotoxic activity	[135]
*Cladophoropsis* sp. (Chlorophyta) (ethanol extract)	MDA-MB-231; MCF-7; T-47D	Growth inhibition due to an estrogen receptor/progesterone receptor-independent mechanism	[136]
*Ulva lactuca* (formerly *Ulva fasciata*) (Chlorophyta) (alcoholic extract)	MCF-7; MDA-MB-231; HeLa; HepG2; HT-29	All the extracts were antiproliferative against all the cancer cell lines, dose-dependently, with the *G. corticata* methanol extract showing the greatest inhibition activity against the MCF-7 cell line	[129]
*Ulva lactuca* (formerly *Ulva fasciata*) (Chlorophyta) (Chloroform extracts)	HepG2; MCF-7; HeLa; PC-3	Strong activity against PC3 and HePG2 cell lines	[137]
*Ulva lactuca* (Chlorophyta) (phycocyanin, methanolic extract)	HepG2; MCF-7	Antiproliferative and pro-apoptotic activities	[138]
*Ulva lactuca* (Chlorophyta) (Chloroform extracts)	HepG2; MCF-7; HeLa; PC-3	Strong activity against MCF-7 and Hela cell lines	[137]
Red Macroalgae			
*Asparagopsis armata* (Rhodophyta) (dichloromethane and methanol extracts)	Caco-2	↓ Cell viability ↓ Cell proliferation	[139]
*Gracilaria corticata* (Rhodophyta) (methanol extract)	MDA-MB 231	Cytotoxic effect	[140]
*Gracilaria corticata* (Rhodophyta) (acqueous extract)	Jurkat; molt-4	Cytotoxic effect of water crude extract	[141]
*Gracillaria corticata* (Rhodophyta) (alcoholic extract)	MCF-7; MDA-MB-231; HeLa; HepG2; HT-29	All the extracts were antiproliferative against all the cancer cell lines, dose-dependently, with the *G. corticata* methanol extract showing the greatest inhibition activity against the MCF-7 cell line	[129]
*Gracilaria foliifera* (Rhodophyta) (ethanol extract)	MDA-MB-231; MCF-7; T-47D	Growth inhibition due to an estrogen receptor/progesterone receptor-independent mechanism	[136]
*Gracilariopsis lemaneiformis* (Rhodophyta) (polysaccharides, water extraction)	A549; MKN28; B16	Inhibition of cell proliferation Induction of apoptosis	[142]
*Iridaea cordata* (Rhodophyta) (ethyl acetate extract)	A-431	High cytotoxic activity	[143]
*Jania rubens* (Rhodophyta) (methanol extract)	HepG2	↓ Number and viability of cells	[127]
*Laurencia obtuse* (Rhodophyta) (Hex:AcOEt fraction)	AGS	Hex:AcOEt fraction of *L. obtusa* was the most cytotoxic against AGS cells	[144]
*Pyropia endiviifolia* (Rhodophyta) (hexan, ethanol and chloroform extract)	U87-MG; HTB-14; A549; CCL-185	Antiproliferative effect of hexane and ethanol extract ↓ Glioma cell with hexane and ethanol extract ↓ Lung adenocarcinoma cell viability with chloroform extract	[117]
*Plocamium cartilagineum* (Rhodophyta) (methan extract)	Caco-2	↓ Cell viability ↓ Cell proliferation	[139]
*Plocamium corallorhiza* (Rhodophyta) (polyhalogenated monoterpene)	MCF-7; MDA-MB-231	Apoptosis induction via caspase-3, -6, -8, -9, and -13 ↓ in Bcl-2 gene Arrest of G1 phase in MDA-MB-231 cells	[128]
*Plocamium cornutum* (Rhodophyta) (polyhalogenated monoterpene)	MCF-7; MDA-MB-231	Apoptosis induction via caspase-3, -6, -8, -9, and -13 ↓ in Bcl-2 gene Arrest of G1 phase in MDA-MB-231 cells	[128]
*Sphaerococcus cornopifolius* (Rhodophyta) (dichloromethane and methanol extracts)	Caco-2	↓ Cell viability ↓ Cell proliferation	[139]

A427: human lung carcinoma; A-431: human epidermoid carcinoma; A549: human lung adenocarcinoma; AGS: human gastric adenocarcinoma; B16: mouse melanoma skin tissue; Caco-2: human colorectal cancer; CCL-185: human lung adenocarcinoma; CL1-5: human lung cancer; Colo357: human pancreas adenocarcinoma; HCC827: human lung adenocarcinoma; HCT116: human colon cancer; HCT-8: human colorectal cancer; HeLa: human cervical cancer; HepG2: human hepatocellular carcinoma; HL-60: human promyelocytic leukemia; HT-29: human colon carcinoma; HTB-14: human glioma; Jurkat: human leukemia; K-562: human leukemia; KAIMRC1: human breast cancer; KG-1a: human leukemia; MCF-7: human breast cancer; MDA-MB-231: human breast cancer; MKN28: stomach adenocarcinoma; molt-4: human leukemia; PANC-1: human pancreatic carcinoma; Panc89: pancreatic cancer; PancTu1: human pancreatic carcinoma; PC-3: human prostate cancer; SK-LU-1: human lung adenocarcinoma; T-47D: human breast cancer; U87-MG: human glioma.

## 6. In Vivo Studies

Given the promising and numerous results obtained in vitro, research focused on the evaluation of the anticancer capacity of algae was also carried out in different in vivo models, of which the murine model is a major representative (Table 3). Currently, in vivo studies reporting the use of pure phytochemicals derived from macroalgae are small due to the variability in the biomass used as a starting point [145]. In a study conducted by Jin et al., rats were injected with the carcinogen diethylnitrosamine (DEN), and the experimental group received fucoxanthin from brown macroalgae in their diet. At the end of the experiment, the incidence of tumors was significantly lower in the group receiving fucoxanthin. During the autopsy, it was further noted that the rats taking fucoxanthin had fewer tumorous characteristics. As a result, the scientists hypothesized that fucoxanthin might have a blocking effect against DEN during both the tumor initiation phase and the promotion phase [146]. In another study, rats were induced with colorectal cancer using a carcinogen and were then given alginates from *Laminaria hyperborea* (Phaeophyceae) as part of their diet. The results demonstrate a significant reduction in the incidence of cancer nodules in the rat intestines, and the reduction in size of these nodules was dose-dependent. Additionally, when the same approach was used on adenoma in the rats’ intestinal tract, the growth of the adenoma was found to be inhibited [147]. Another study conducted by Hwang et al. disclosed that brown algae polyphenols, administered both orally and topically, significantly reduced UVB-induced skin tumor multiplicity and volume in SKH-1 mice, likely by inhibiting oxidative stress and inflammation [148]. Lastly, eckol, a phlorotannin from brown algae, exhibited potent antitumor effects in sarcoma 180 xenograft-bearing mice by promoting apoptosis, inhibiting cell proliferation, and enhancing immune responses, including increased T lymphocyte activity and dendritic cell infiltration [149]. Concurrent with studies that have seen the administration of extracts of bioactive molecules, several studies report the use of whole algae within the diet. In rats, it was observed that supplementation of the brown algae *Dictyota dichotoma* (Phaeophyceae) in the diet was well-tolerated at high doses, indicating its potential for safe anticancer applications [150]. Different studies were also conducted in mice models. In this context, the administration of extracts from the brown alga *Padina pavonica* (Phaeophyceae) and the red alga *Jania rubens* discloses the ability to reduce the tumor cell number in vivo and enhance the immune responses in mice challenged with Ehrlich ascites carcinoma cells [127]. The administration of seaweed makes it possible to evaluate the synergistic effect of the algal phytocomplexes, which could lead to a greater antitumor effect, as the bioactive compounds working synergistically are able to reduce the cytotoxic effects that the administration of a single concentrated compound could cause not only toward cancer cells but also toward healthy ones [151]. In addition to this aspect, the administration of algal biomass in its entirety meets the recommendations of the guidelines in the field of precision and functional nutrition as a key element in cancer treatment [152,153].

Collectively, these studies underscore the diverse and potent anticancer properties of various algae. They not only validate the potential of algae as therapeutic agents but also highlight their role as dietary supplements in cancer prevention and treatment. This dual role of algae in combating cancer is a significant finding of our research.

## 7. Potential Applications in Cancer Treatment

To date, the most effective method of treating cancer is the use of toxic chemicals that preferentially kill cancer cells. Unfortunately, these chemical drugs lack selectivity and affect normal cells, causing a number of side effects, including damage to the immune system and organs, such as the liver and kidneys. This often leads to reducing the dosage of the therapy so that the side effects are tolerable [157,158]. In recent years, there has been a growing interest in the study of using algae in cancer treatment. This is largely due to the observations that many types of algae have substantially different effects when tested on normal cells compared to cancerous cells. This selectiveness for cancer cells is the desired characteristic of an anticancer agent. Therefore, the hypothesis currently being tested identifies algae as a potential adjuvant in therapies to support either the synthetic or natural anticancer drugs that are currently being used. This may improve their effectiveness by acting as a sensitizer, or it may allow a lower dosage of the drug to be used [159,160].

In the context of combination therapy, it is now possible to utilize antioxidants and anticancer agents derived from algae to enhance the effectiveness of standard chemotherapeutic agents. By combining an anticancer agent with an antioxidant, it is potentially possible to decrease the toxic side effects of chemotherapy on healthy cells [161,162]. There are various types of combination therapy available to fight different forms of cancer. For instance, research has demonstrated that the administration of vitamin E and vitamin C can aid the anticancer activity of certain cytotoxic agents. This is because these vitamins can reduce the amount of cytotoxic destruction caused by free radicals on normal cells [163,164,165,166]. Moreover, the use of PUFA derived from algae has the potential to enhance the cytotoxic effects of chemotherapeutic agents on hormone-dependent cancers. Recent research has, in fact, demonstrated that combining PUFA with tamoxifen can effectively induce apoptosis in cancer cells. This may be attributed to the ability of PUFA to increase the expression of estrogen receptors, which has been associated with a higher susceptibility of hormone-dependent cancers to tamoxifen [167,168]. Additionally, PUFAs are known to be highly effective in inhibiting the synthesis of cyclooxygenase enzymes and PGE2, both of which have been implicated in tumor progression [169,170,171].

The success of drug delivery depends on the carrier system used. The carrier must ensure that the therapeutic agent remains undisturbed until it reaches its target site, and once released, it should not be metabolized or excreted prematurely [172,173,174]. Therefore, researchers have shown interest in utilizing macrophages to deliver anticancer drugs to tumors. In fact, macrophages have the ability to migrate and penetrate solid tumors, as well as express certain receptors absent or present in low concentrations on non-malignant cells. One such example is suicide gene therapy for metastatic melanoma and other metastatic cancers, employing genetically modified human CD14+ peripheral blood macrophages. These macrophages carry the tumor migration inhibitory factor (TMIG) gene under a heat-inducible promoter and are encapsulated in alginate microcapsules containing low-cost synthetic cis aconitate-based fluorinated anti-metabolites with antitumor properties. By systemically delivering these TMIG gene macrophages to metastatic tumor sites through heat induction, the macrophages revert their phenotype from migratory to non-migratory, effectively trapping them at the tumor site and maximizing the antitumor effect [62,175]. The field of gene therapy presents an opportunity for macro- and microalgae to be utilized in targeted drug delivery.

## 8. Side Effects

Exploring macroalgae as potential antitumor agents has gained significant attention due to their diverse bioactivities and the unique compounds they produce. However, like any therapeutic intervention, these natural compounds are not without their risks. The side effects and potential toxicities of macroalgae-derived treatments are influenced by the specific bioactive compounds present, their mechanisms of action, and the administered dosage. As reported in the preceding paragraphs, some macroalgae-derived compounds can exert a cytotoxic effect against cancer cells, but, at the same time, they might demonstrate the same effect against cells of healthy tissues. This is, for example, the case found by Alves and colleagues, where extracts from *Asparagopsis armata* and *Sphaerococcus coronopifolius* demonstrated potent cytotoxic activity against colorectal cancer cells but without precise targeting, so these effects could extend to normal cells, leading to unintended toxicity [139]. In addition, some macroalgae-derived metabolites, particularly brominated compounds found in red algae, exhibit broad-spectrum activity that does not always differentiate between cancerous and healthy cells, thus necessitating the need for appropriate targeting [176]. Additional to the direct toxicity that can be caused by algae compounds, it is known that the pro-apoptotic properties of algal extracts are often mediated by the induction of oxidative stress, which facilitates the destruction of malignant cells. However, this same mechanism can adversely impact normal cells, leading to neurotoxicity and broader cytotoxic effects. Research on MCF-7 breast cancer cells suggests that oxidative stress induced by certain algal compounds can be a double-edged sword, requiring careful modulation to minimize collateral damage [177]. Moreover, the use of algae in the treatment of cancer can lead to modulation of the immune system. In this context, certain algae-derived compounds have immunomodulatory properties that can either enhance or suppress immune system activity. While some polysaccharides stimulate immune responses to target tumors, others may inadvertently trigger excessive inflammation or immune suppression, leading to unintended consequences such as autoimmune reactions or increased susceptibility to infections [178]. The identification of the aforementioned adverse effects is primarily attributable to the fact that the exact correlation between the chemical structure of macroalgae-derived compounds and their biological activity is not yet fully understood [176]. This lack of knowledge makes it difficult to predict, control, and mitigate the potential toxicities associated with these natural products. Furthermore, although numerous in vitro and in vivo studies highlight the promising antitumor properties of macroalgae, clinical studies remain scarce, showing significant variations in both antitumor efficacy and toxicity among different species and even among individuals of the same species [119].

The side effects reported to date may be further reduced in the coming years, as a deeper understanding of the molecular mechanisms underlying the antitumor effects of macroalgae—including their ability to promote apoptosis and inhibit angiogenesis—can aid in designing more selective treatments. At the same time, certain macroalgae-derived compounds, such as phlorotannins, possess antioxidant properties that can counteract oxidative stress-induced damage. This contributes to mitigating the side effects associated with oxidative toxicity, providing protection to non-cancerous cells [22]. By refining these mechanisms, researchers could develop therapies that minimize harm to healthy cells while maximizing antitumor efficacy [97,119].

## 9. Challenges and Future Directions

Algal compounds have shown potential in combating cancer, but further research is needed before clinical trials can be conducted. Prior to clinical trials, a preclinical pathway must be followed to evaluate the effectiveness of algal compounds for cancer treatment. This pathway will include (i) standardizing algal extracts chemically and biologically; (ii) conducting toxicity studies using normal cells and animal models; and (iii) investigating the mechanisms of action of the algal compounds [160,179,180,181].

As mentioned above, the compounds with anticancer activity derived from macroalgae comprise intricate mixtures primarily consisting of polysaccharides, proteins, lipids, pigments, vitamins, and minerals. Therefore, it is necessary to find for each compound the best extraction method since the efficacy of the extraction process, as well as the chosen solvent, heavily influences the quantity and quality of the algal extracts, which ultimately impacts their biological functions. To ensure reliable findings in forthcoming research, it is imperative to establish a standardized technique for extracting algal compounds [182,183,184,185,186]. Although various methods exist for this purpose, there is no universally accepted approach. Nevertheless, the National Cancer Institute (NCI) method has been widely employed for standardizing extracts. This bioassay involves diluting the extract to determine its IC50 value, which indicates its ability to inhibit cancer cell growth. If the extract exhibits anticancer activity by preventing cell proliferation, an IC_50_ value of less than 20 μg/mL is obtained. Even if this approach is not official, the information obtained from this assay is valuable as it allows for comparisons of an extract’s potency over time and across different research groups [187]. Once the extraction is complete, the identification and isolation of particular compounds will allow further evaluation of these pure compounds’ use in vitro and in vivo cancer models. Despite the benefits, there are drawbacks to consider, including the need for substantial amounts of extract, the lack of correlation to in vivo activity, and the associated cost [188].

Moreover, another crucial point that has to be investigated before proceeding with clinical trials is formulation development. This could involve simply administering extracts in the form of capsules or taking more complex steps to develop the extracts into functional foods or nutraceuticals. Preliminary investigations are conducted to evaluate the physical and chemical characteristics of medication and the viability of creating a specific drug form. When it comes to natural substances, there is limited knowledge regarding the primary elements found in extracts. Nevertheless, in certain cases where the active ingredient has been isolated, additional spectroscopic examinations can be conducted to determine the compound’s structure [189,190,191,192]. Subsequently, the design of the formulation and the optimization of the product can be carried out, which can vary in complexity based on the nature of the extract and the intended form of administration.

## 10. Conclusions

This review provides an overview of the anticancer properties of extracts derived from macroalgae. Research in recent years has highlighted the potential of several algae-derived compounds, such as pigments, polyphenols, polysaccharides, proteins, and peptides, to inhibit cancer cell growth and proliferation, induce apoptosis (programmed cell death), modulate the immune system, and reduce side effects associated with traditional therapies. Algae represent a renewable and sustainable source of bioactive compounds with high potential for the development of new anticancer therapies. Their multifunctional nature and low toxicity make them ideal candidates for combination with traditional chemotherapeutic drugs, enhancing their efficacy and reducing their side effects. However, it is important to note that research in this field is still in its early stages, and further clinical trials are needed to confirm the efficacy and safety of algae in cancer therapy. Nevertheless, their prospects are promising, and algae represent real hope for the development of new, more effective, and less invasive cancer therapies.

## Figures and Tables

**Figure 1 marinedrugs-23-00070-f001:**
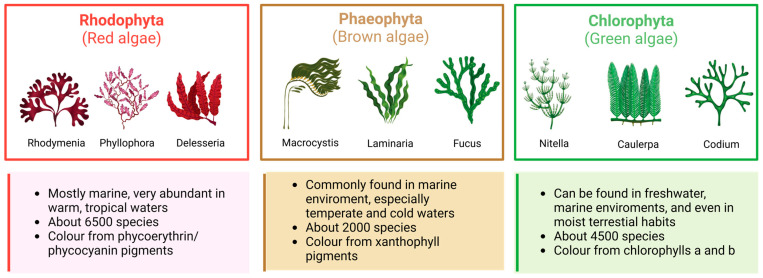
Classification of macroalgae and their main characteristics: Rhodophyta (red algae); Phaeophyta (brown algae); and Chlorophyta (green algae).

**Figure 2 marinedrugs-23-00070-f002:**
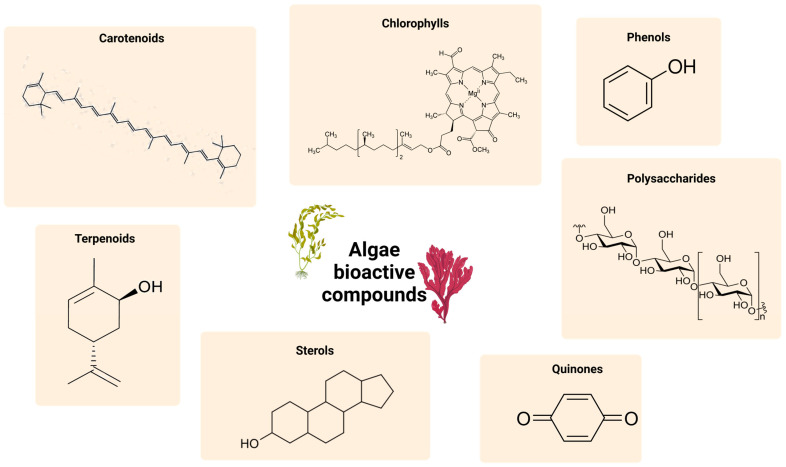
Representation of the molecular structure of the main compound classes of algae-derived bioactive compounds.

**Figure 3 marinedrugs-23-00070-f003:**
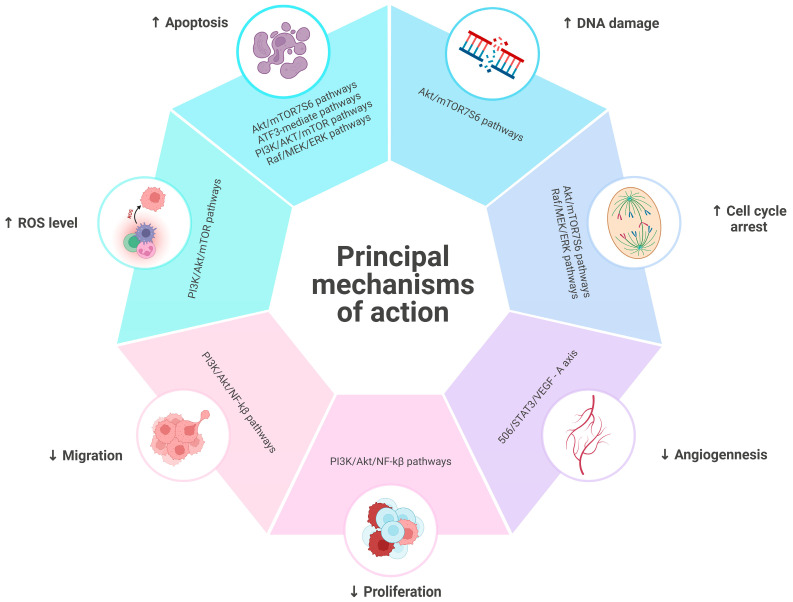
Representation of the main mechanisms of action regulating the antitumor effect of algae-derived bioactive compounds.

**Table 1 marinedrugs-23-00070-t001:** Main classes of bioactive compounds and relative chemical compounds in algae species.

Compound Class	Example Compounds	Algae Species	Reference
Carotenoids	Alloxantin	*Cryptomonas ovata* (Cryptophyceae) *Cryptomonas erosa* (Cryptophyceae) *Rhodomonas salina* (Cryptophyceae)	[55,56,57,58]
	Crocoxanthin	*Karenia brevis* (Dinoflagellata)	
	Fucoxanthin	*Undaria pinnatifida* (Phaeophyceae) *Phaeodactylum tricornutum* (Bacillariophycae)	
	Fucoxanthinol	*Undaria pinnatifida* (Phaeophyceae) *Saccharina japonica* (formerly *Laminaria japonica*) (Phaeophyceae) *Sargassum* spp. (Phaeophyceae)	
	Siphonaxanthin	*Codium fragile* (Chlorophyta) *Caulerpa lentillifera* (Chlorophyta) *Umbraulva japonica* (Chlorophyta)	
	Xanthophylls	*Undaria pinnatifida* (Phaeophyceae) *Laminaria japonica* (formerly *Laminaria japonica*) (Phaeophyceae) *Sargassum* spp. (Phaeophyceae)	
Chlorophylls	Chlorophyll a and b	*Ulva lactuca* (Chlorophyta) *Codium fragile* (Chlorophyta) *Caulerpa lentillifera* (Chlorophyta)	[59,60,61]
Phenols	2-Bromophenol	*Rhodomela larix* (formerly *Rhodomela larix*) (Rhodophyta)	[62,63]
	Phloroglucinol	*Ecklonia cava* (Phaeophyceae) *Sargassum* spp. (Phaeophyceae)	
	Phlorotannins	*Himanthalia elongata* (Phaeophyceae) *Halopteris scoparia* (formerly *Stypocaulon scoparium*) (Phaeophyceae) *Ascophyllum nodosum* (Phaeophyceae) *Ecklonia cava* subsp. stolonifera (formerly *Ecklonia stolonifera*) (Phaeophyceae) *Fucus vesiculosus* (Phaeophyceae) *Macrocystis pyrifera* (Phaeophyceae)	
Polysaccharides	Alginate	*Laminaria digitata* (Phaeophyceae) *Laminaria hyperborea* (Phaeophyceae) *Macrocystis pyrifera* (Phaeophyceae) *Ascophyllum nodosum* (Phaeophyceae)	[64,65,66,67]
	Carrageenan	*Kappaphycus alvarezii* (Rhodophyta) *Eucheuma denticulatum* (Rhodophyta)	
	Fucoidan	*Fucus vesiculosus* (Phaeophyceae) *Laminaria* spp. (Phaeophyceae)	
	Laminarin	*Saccharina latissima* (Phaeophyceae) *Laminaria* spp. (Phaeophyceae)	
Quinones	Hydroquinone	*Sargassum polycystum* (Phaeophyceae)	[68,69]
	Quinone	*Sargassum* spp. (Phaeophyceae)	
Sterols	Fucosterol	*Pelvetia canaliculata* (Phaeophyceae) *Fucus vesiculosus* (Phaeophyceae)	[70,71]
	Phytosterol	*Codium tomentosum* (Chlorophyta) *Saccharina latissima* (Phaeophyceae) *Gracilaria* spp. (Rhodophyta)	
Terpenoids	Diterpenoid	*Dictyota* spp. (Phaeophyceae) *Bifurcaria bifurcata* (Phaeophyceae)	[72,73,74,75,76]
	Monoterpene	*Portieria hornemannii* (Rhodophyta)	
	Triterpenoid	*Gracilaria salicornia* (Rhodophyta) *Padina boergesenii* (Phaeophyceae)	

**Table 3 marinedrugs-23-00070-t003:** In vivo studies reporting the anticancer potential of brown, red, and green macroalgae.

Algae Species	Animal Model	Experimental Group	Effect	Reference
Brown algae polyphenols	Female SKH-1 hairless mice (7–8 weeks of age)	Oral treatment groups: (i) control; (ii) UVB control; (iii) UVB + brown algae polyphenols. Topical treatment groups: (i) solvent vehicle control; brown algae polyphenol control; UVB + solvent vehicle control; UVB + brown algae polyphenols.	Dietary feeding: - ↓ Tumor multiplicity (45% and 56%); - ↓ Tumor volume (54% and 65%). Topical administration (3 and 6 mg): - ↓ Tumor multiplicity (60% and 46%); - ↓ Tumor volume (66% and 57%). Inhibition of cyclooxygenase-2 activity and cell proliferation.	[148]
*Ericaria selaginoides* (formerly *Cystoseira tamariscifolia*) (Phaeophyceae)	Swiss albinos mice (6–8 weeks of age)	Lot 1 (1250 mg/kg); Lot 2 (1041.66 mg/kg); Lot 3 (868.05 mg/kg); Lot 4 (723.27 mg/kg); Lot 5 (602.81 mg/kg); Lot 6 (502.34 mg/kg); Lot 7 (0 mg/kg).	Toxic effect of the alga on the mice with LD_50_ of 738.61 ± 34 µg of alga/g of body weight.	[154]
*Chlorella vulgaris* (Chlorophyta)	Female Balb/c mice and nude mice	(1) Control; (2) Laser; (3) RT; (4) Laser + RT; (5) Algae@SiO_2_; (6) Algae@SiO_2_ + laser; (7) Algae@SiO_2_ + RT; (8) Algae@SiO_2_ + laser + RT.	- Algae@SiO_2_ ↑ oxygen production capability that alleviates the tumor hypoxia; - With X-ray irradiation, proliferation and mitigation of 4T1 cells were suppressed; - Inhibition of tumor development with laser and RT treatment for the Algae@SiO_2_-mediated combination therapy; - RT and PDT combination therapy inhibit the pulmonary metastasis of the 4T1 breast tumor.	[155]
Eckol derived from marine brown algae	Male Kunming mice	(1) Control; (2) Eckol low dose (0.25 mg kg^−1^); (3) Eckol middle dose (0.5 mg kg^−1^); (4) Eckol high dose (1.0 mg kg^−1^).	- ↑ TUNEL-positive apoptotic cells; - ↑ Caspase-3 and Caspase-9 expression; - ↓ Bcl-2, Bax, EGFR, and p-EGFR expression stimulation of mononuclear phagocytic system. ↑ CD4+/CD8+ T lymphocyte ratio; - ↑ Cytotoxic T lymphocyte responses.	[149]
Pandina pavonica (Phaeophyceae) Jania rubens (Rhodophyta)	Female Swiss albino mice CD1 (6–8 weeks old)	(1) Naïve EAC control (i.p. saline solution after EAC transplantations); (2) Positive EAC control (10 µg Cisplatin^®^/mouse i.p. after EAC transplantations); (3) *P. pavonica* extract (2.5 μg/mouse i.p. after EAC transplantations); (4) *P. pavonica* extract (1.3 μg/mouse i.p. after EAC transplantations); (5) *J. rubens* extract (2.3 μg/mouse i.p. after EAC transplantations); (6) *J. rubens* extract (1.2 μg/mouse i.p. after EAC transplantations); (7) *P. pavonica* extract (2.5, μg/mouse i.p. before EAC transplantations); (8) *J. rubens* extracts (2.3 μg/mouse i.p. before EAC transplantations).	- ↓ Number and viability of EAC tumor cells; - ↑ EAC apoptosis.	[127]
Dimethylsulfoniopropionate (DMSP) of green algae	Male ICR/Jcl mice (4 weeks old)	(1) Control; (2) EAC control; (3) DMSP (5 mM) + EAC; (4) DMSP (10 mM) + EAC; (5) DMPS (20 mM) + EAC.	- 10 and 20 mM DMSP solutions prolonged the lives span of mice; - 10 mM DMSP solution activated delayed-type hypersensitivity.	[156]

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
