# Peer review of "Anticancer Properties of Macroalgae: A Comprehensive Review"

_marinedrugs, 2025, doi:10.3390/md23020070_

Round 1

Reviewer 1 Report

Comments and Suggestions for Authors

In marinedrugs-3466009, Sara Frazzini  and Luciana Rossi discuss the anticancer properties and potentials of macro-algae. The topic of this review is interesting and fits well the scope of Marinedrugs. The reviewer feels this manuscript need to be amended extensively before it can be published. 

(1) The title is miss-leading. Anticancer effects include anticarcinogenic effects. The title must be changed.

(2) As a review, the presentation is very important. Colorful figure is crucial. However, the authors' presentation is really so so.

(3) It is meaningless to discuss the anticancer effect of the extract in cell culture model. Only in vivo studies carred out with pure phyto-chemicals are really clinically relevant. It is appropriate to discuss the in vivo effects of raw extracts in animal models as they are clinically relevant.

(4) Any side effect and potential toxicity? Such information should be discussed as well.

Anticancer and Anticarcinogenic Properties of Macro-algae: A 2 Comprehensive Review 3

Author Response

Dear reviewer, we thank you for taking the time to review our manuscript. Attached you will find the point-by-point responses and the changes in the text are in red color

Reviewer 2 Report

Comments and Suggestions for Authors

Manuscript “Anticancer and Anticarcinogenic Properties of Macro-algae: A Comprehensive Review” is a review in the field of application of compounds isolated from Macro-algae.

Cancer indeed is a very urgent and not only medicinal but social problem of our civilization. In this case the search for anti-cancer agents in natural sources is a pressing issue. Such review on the presence and isolation of such substances from macro-algae is relevant and timely. This topic is well suited to the Marine Drugs Journal and will be of interest to readers.

Manuscript is based on 169 literature references most of which have been published in the last 5 years.

Manuscript consists of 9 sections Including introduction and Conclusions. The sections are the following: 2. Macro-algae: A Comprehensive Overview; 3. Macroalgae's bioactive compounds with anticancer and anticarcinogenic potential; 4. Mechanisms of Action; 5. In vitro Studies; 6. In vivo Studies; 7. Potential Applications in Cancer Treatment; 8. Challenges and Future Directions.

1.       The main issue of the manuscript is about the difference between Anticancer and Anticarcinogenic. In the title you highlight Anticancer and Anticarcinogenic. However through the manuscript there is no explanation of the difference and no division. You can separate your compounds in table 1, 2 and 3 whether they possess Anticancer or Anticarcinogenic effect. The same in section 4. It starts with “The anticancer effect…” but no further data is given about Anticarcinogenic effect. Conclusions also starting with “This review provides an overview of the anti-cancer properties of…” and has nothing about Anticarcinogenic properties.

2.       Add title for Table 1

3.       There is “Anticancer” in the title and Section 4, but “anti-cancer” in the conclusions. Please check through the manuscript and make it consistent.

Author Response

(The authors gave the same response as above.)

Reviewer 3 Report

Comments and Suggestions for Authors

In my opinion, this is a good, interesting, quality and comprehensive work. The work is written in clear language and well structured and I believe that this work will be of interest to readers of the Marine Drugs journal especially to the Special Issue “Marine Algae: Exploring Their Nutritional, Health, and Nutraceutical Potential”.

However, it is recommended to make the following corrections and additions before publication:

1) Table 1 does not look like a table at all. It is rather a drawing, where the structures of biologically active compounds are presented by classes of these compounds (carotenoids, phenols, polysaccharides, terpenes, etc.). However, there is no explanation of which specific algae species were used to isolate and characterize specific chemical structures. It is also unclear whether all classes of compounds are found in algae or whether different classes of compounds are characteristic of specific representatives. References to literature are also needed. Therefore, I would advise the authors to present this undoubtedly important material in the form of a table, where there will be additional columns with all the necessary information. As for technical comments on this table, these are: 1) there is no caption to the table (there is only a link in the text) 2) All structures of the compounds are depicted in poor quality. The size of the atoms looks too small.

2) Section 3 mentions the bioactive substances siphonaxanthin, lutein, zeaxanthin, β-carotene and others, the structures of which are not presented in Table 1. If the purpose of Table 1 is simply to illustrate the diversity of chemical structures and classes of compounds in algae, then it is unclear why the table does not include those structures that are further described in the text, with references to the literature. And vice versa, why is a description not added for the structures presented in the table? At least selectively.

3) Figure 1. Although the design of the figure looks very good, completely upside-down text is not the best idea.

4) In the test of section 4 there is no mention of some mechanisms such as ATF-3-mediate pathway, Raf/MEK/ERK pathway, presented in Figure 1.

5) Line 204. The phrase "studies have revealed that algae-derived compounds can modulate the tumor microenvironment". What compounds do the authors mean?

6) Line 213. The phrase "these compounds have demonstrated the potential to sensitize cancer cells to radiation therapy." Same question. What compounds do the authors mean?

7) In Table 2, in those cases where not specific compounds but simply extracts are described, it is necessary, if such data is available, to describe in more detail the composition of the extract, what compounds are included.

8) Line 230. The word toxin is not correct for antitumor compounds.

Author Response

(The authors gave the same response as above.)

Round 2

Reviewer 2 Report

Comments and Suggestions for Authors

Manuscript could be accepted.

Author Response

Dear reviewer, we thank you for taking the time to review our manuscript.